# CBNet: Cooperation-Based Weakly Supervised Polyp Detection

Xiuquan Du*†
Key Laboratory of Intelligent
Computing and Signal Processing of
Ministry of Education
School of Computer Science and
Technology, Anhui University
Hefei, China
dxqllp@ahu.edu.cn

Jiajia Chen*
School of Computer Science and
Technology, Anhui University
Hefei, China
e22301271@stu.ahu.edu.cn

Xuejun Zhang
School of Computer Science and
Technology, Anhui University
Hefei, China
e22201060@stu.ahu.edu.cn

## Abstract

Missed polyps are the major risk factor for colorectal cancer. To minimize misdiagnosis, many methods have been developed. However, they either rely on laborious instance-level annotations, require labeling of prompt points, or lack the ability to filter noise proposals and detect polyps integrally, resulting in severe challenges in this area. In this paper, we propose a novel *C*ooperation-*B*ased network (***CBNet***), a two-stage polyp detection framework supervised by image labels that removes wrong proposals through classification in collaboration with segmentation and obtains a more accurate detector by aggregating adaptive multi-level regional features. Specifically, we conduct a Cooperation-Based Region Proposal Network (CBRPN) to reduce the negative impact of noises by deleting proposals without polyps, enabling our network to capture polyp features. Moreover, to enhance location integrity and classification precision of polyps, we aggregate multi-level region of interest (ROI) features under the guidance of the backbone classification layer, namely Adaptive ROI Fusion Module (ARFM). Extensive experiments on the public and private datasets show that our method achieves state-of-the-art performance for weakly supervised methods and even outperforms full supervision in some terms. All code is available at ***https://github.com/dxqllp/CBNet***.

## CCS Concepts

• **Computing methodologies → Object detection**.

## Keywords

Polyps, Cooperation, Weakly Supervised Detection

**ACM Reference Format:**
Xiuquan Du, Jiajia Chen, and Xuejun Zhang. 2024. CBNet: Cooperation-Based Weakly Supervised Polyp Detection. In *Proceedings of the 32nd ACM International Conference on Multimedia (MM '24), October 28-November 1,* 2024, *Melbourne, VIC, Australia.* ACM, New York, NY, USA, 9 pages. https://doi.org/10.1145/3664647.3680991

---

*Both authors contributed equally to this work.
†Corresponding author

## 1 Introduction

Colorectal cancer (CRC) ranks as the third most frequent cancer, with more than 80% originating from polyps [31]. However, untreated polyps might become malignant and life-threatening cancer [22]. In clinical practice, the colonoscopy procedure is regarded as the golden standard for detecting and removing polyps. Unfortunately, the method highly depends on the experience and proficiency of the medical practitioner that suffers a high miss rate (as much as 30%) [37]. Fortunately, computer-aided detection (CAD) has been proven to help doctors detect polyps and reduce the incidence of missed diagnoses [20]. Although CAD has demonstrated

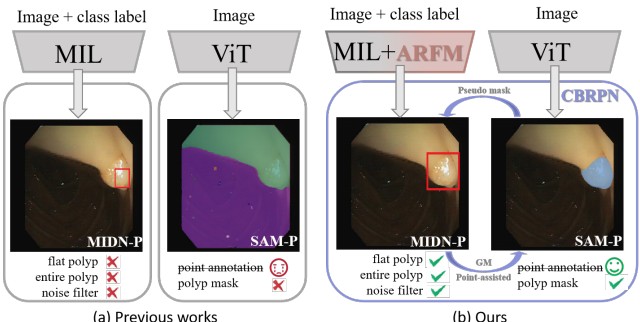

**Figure 1: The schematic diagram of the previous works with separable mechanism and our proposed collaboration approach. And four extremely challenging scenarios, i.e., missed flat polyp, non-entire polyp, low-quality proposals with unfilterable noise and wrong polyp mask with point annotations existed in polyp detection.**

impressive capabilities in polyp detection [4, 15, 19, 24, 26, 33, 36], they need to be combined with instance-level bounding box annotations, which are very laborious and challenging owing to the complexity of colonic images and the diversity of polyps. To reduce the labeling burden, researchers hope to make detectors in a weakly supervised (WS) fashion. For example, several promising works combining multiple instance learning (MIL) with deep learning [21, 25, 39] have greatly pushed the boundaries of natural images to successfully apply WS to the medical field. Unfortunately, accurate and reliable polyp detection can be easily fooled by **flat polyps** because they lack clearly visible borders and exhibit a similar aspect

to the surrounding colorectum, which makes the network struggle to accurately distinguish polyps from backgrounds. In addition, most of these studies directly use coarse proposals generated by standard methods for training, yet most of them are negative cases (**noise**) that do not contain the target, which affects the learning of the target features while lowering the performance of the detection.

Worse still, they severely rely on region-level classifiers that overly focus on the most discriminatory local regions (**over-fitting**), which not only reduces classification accuracy but also results in over-fitting, as shown in Figure 1 (a) left. More recently, some high-profile works hope to solve this task through transfer learning, such as weakly supervised polyp segmentation (WSPS) with the help of the Segment Anything Model (SAM). But it is essentially a promotable method, which **needs point annotations** and will lose the ability to accurately identify polyps if there are no additional point annotations as shown in Figure 1 (a) right. Additional points not only increase the burden of annotation but also deviate from the requirements of only image labels. Therefore, a further mechanism is necessary for a more reasonable inference.

Through deeper study, we find that the MIL-based and SAM-aided approaches have complementary advantages and disadvantages. For instance, the classifiers among MIL tend to focus on parts of objects that match the need for prompts in SAM, because it can automatically generate point prompts based on the focused region, and eliminate the burden for additional point annotations. Meanwhile, the high-quality polyp masks produced from point prompts can not only ensure the integrity of the target to avoid over-fitting but also filter the noise in the proposal so that the network can better learn polyp features and gain inspiring performance. In a word, instead of working separately, they should cooperate with each other to overcome their internal weaknesses.

As a result, in this work, we innovatively come up with a new solution, namely CBNet, as shown in Figure 1 (b), a weakly supervised polyp detection model is designed to solve the above challenges: (1) over-fitting polyp detection. (2) unfilterable noise. (3) failed polyp mask without point annotation. (4) missed flat polyps. Our approach mainly includes two innovative modules: Cooperation-Based Region Proposal Network (CBRPN), Adaptive ROI Fusion Module (ARFM), and a common module: Multiple Instance Detection Network (MIDN). Specifically, inspired by the complementary strengths and weaknesses, we design **CBRPN** mainly for **challenges (2) - (3)** while initially addressing **challenge (1)**, which includes a proposal generator and SAM. The former generates a series of rough proposals by graph-based segmentation and merging strategies. The latter generates pseudo masks based on point prompts (derived from the gradient matrix of the backbone). After that masks and proposals are computed intersection over union (IOU) one by one to leave proposals that contain relatively full polyps and remove the noise without polyps. Furthermore, we employ **ARFM** to solve **challenge (4)** and further address **challenge (1)**, which consists of two parts: classification layer and ROI fusion. During the training of the module, the parameters of the classification layer are frozen to ensure that the backbone network can focus on polyps from the whole image for further completeness, and judge the morphology of them from the global features to improve classification accuracy. In the fusion part, the differences between polyps and background will be refined by combining deep-shallow region features to improve

the detection of flat polyps. This module adaptively adjusts features learned in the second stage, resulting in a more comprehensive global-local feature. Finally, we introduce MIDN for setting the target existence and category possibility scores of the proposal region to achieve weakly supervised detection under the supervision of image-level labels.

In summary, our main contributions are listed four-fold:

- We give a new solution under the weakly supervised, CBNet, a framework that employs the collaborative mechanism to detect polyps with only image-level annotation.
- In the proposals generation stage, we confirm the collaborative nature of the classifier and SAM. Accordingly, we conduct a cooperation-based region proposal network, which implements proposal noise filtering to ensure the correctness of feature extraction, and removes the proposal with incomplete polyp to reduce the risk of over-fitting.
- To better capture the flat polyps and further ensure positional integrity, we design the adaptive ROI fusion module to learn polyp features from the global-local level and refine slight disparities between polyps & background from deep-shallow level.
- Our method is conducted on three datasets (i.e. CVC-ClinicDB, Kvasir, private), which not only obtains state-of-the-art performance in weakly supervised methods but also exceeds fully supervised performance in some respects.

## 2 Related Work

### 2.1 Region Proposals Generation

Region proposal generation methods could be categorized into traditional methods and CNN-based ones where the former category can be further divided into two approaches, i.e., edge-based [8] and superpixel-based [14, 16]. Edge-based methods evaluate window boxes by image edges to refine their location, which struggles in dealing with boundary-blurred polyps. Superpixel-based methods such as Selective Search (SS), can adapt to polyps with blurred boundaries but have a high false alarm rate in complex polyps with various variations. And due to the similarity between polyp and background, the locality of these methods leads to high false alarms and low detection rates. CNN-based methods such as Region Proposal Network (RPN) [1, 28] have been proposed to optimize and filter proposals for faster detection and made great progress in this area. Nevertheless, to ensure high performance, it requires bounding box annotations and sets a large number of hyperparameters for training, which deviates from the weak supervision requirement that only image-level annotations are available (see **supplementary material** for more information). Different from them, we explore new ways to generate proposals and propose a novel CBRPN that can take advantage of both SS and RPN and show a better result in the same network.

### 2.2 Weakly Supervised Cooperation Detection

Due to the absence of instance-level annotations, weakly supervised detection methods are easy to over-fit on object parts. To address this issue, many cooperation-based works such as C-MIDN [12], P-MIDN [38], WS-JDS [30] and NDI-WSOD [35], have been proposed. Despite achieving promising results, over-fitting is still a

---

**Algorithm 1** The first (1) & second (2) stages training

---

**Input:** 1: training set with polyp image label $\mathbf{T_1} = \{(\mathbf{I}, \mathbf{y})\}$.

1:     forward CBNet: $backbone(\mathbf{I}) \rightarrow \{F_5, X^{img}, (x_{point}, y_{point})\}$.

2:     generate proposals sets: $SAM(I, (x_{point}, y_{point})) \rightarrow B_{SAM}$, $SSW(\mathbf{I}) \rightarrow B_{SSW}$.

3:     generate scores: $Softmax(X^{img}) \rightarrow S^{img}$.

4:     generate refined proposals set: ITF $(B_{SSW}, B_{SAM}) \rightarrow B_{CBRPN}$.

5:     compute and backward $L_{img}$ in Eq. 12 for CBNet.

6:     continue until convergence.

**Output:** 1: fixed parameters $Conv\_P$; refined proposals $B_{CBRPN}$.

**Input:** 2: training set $\mathbf{T_2} = \{(\mathbf{I}, B_{CBRPN}, \mathbf{y})\}$ and fixed parameters $Conv\_P$.

1:     forward CBNet: $backbone(\mathbf{I}) \rightarrow \{\{F_1, F_2, F_3, F_4, F_5\}, X^{img}\}$.

2:     generate aggregated feature maps: $Agg(F_i) \rightarrow \hat{F}_i$; crop roi $(R_i)$ according to $B_{CBRPN}$.

3:     generate fusion feature maps: $fusion(R_i) \rightarrow \{X_R^{fusion}, X_C^{fusion}\}$.

4:     generate image feature maps: $GAP(Conv\_P(F_5))$.

5:     generate scores: $Softmax(X^{img}) \rightarrow S^{img}, Softmax_C(X^{img}) \rightarrow S_C^{MI}, Softmax_R(X^{img}) \rightarrow S_R^{MI}$.

6:     combine region score to image scores according to Eq. 10.

7:     compute and backward $L_{img} + L_{MI}$ in Eq. 12 for CBNet.

8:     continue until convergence

**Output:** 2: the optimized CBNet for polyp detection.

---

challenging task because segmentation networks in collaboration are explicitly tailored for the specific domain and their performance can degrade significantly when applied to different types of imaging data. Recently, SAM attracted a lot of attention and was introduced into the medical area [17, 19, 41], owing to the excellent ability of generality. However, it is still unexplored to employ SAM in weakly supervised polyp segmentation only with image annotations. Towards the same goal, we propose the novel CBRPN reduces the possibility of over-fitting and successfully employs SAM for image-supervised WSPS. Meanwhile, ARFM is also designed to further avoid over-fitting while better detecting flat polyps. Both of them have less complexity and particularity while performing better.

## 3 Method

### 3.1 Overview

With the aim of high-quality proposals under image annotations and more accurate detection, we designed CBNet. The overall architecture of our network is shown in Figure 2. The proposed CBNet consists of two stages during training and testing, where the first stage is the region proposal network (CBRPN, see Section 3.3) for proposal generation and the second stage is the WS network (ARFM, see Section 3.4 & MIDN, see Section 3.5) of polyp detection. For more clarity, the training and the testing of our CBNet are summarized in Algorithm 1.

**First Stage:** Given input image $\mathbf{I} \in \mathcal{R}^{C \times H \times W}$ (C: channels, H: height, W: width), it is simultaneously fed into SSW (Selective Search Windows), pre-trained backbone (see Section 3.1) and SAM for generating a coarse set of window boxes $B_{SSW}$, obtaining the

most significant point in the polyp, and producing the other box set $B_{SAM}$ (according to the pseudo-mask) for filtering, respectively. After that, $B_{SSW}$ and $B_{SAM}$ will be sent to iou threshold filter (ITF) to remove noise, and the retained proposals form $B_{CBRPN}$.

**Second Stage:** Features $F_1$-$F_5$ from the backbone and refined proposals $B_{CBRPN}$ from the first stage are processed by ARFM, which contains three cross-layer aggregation blocks stacked sequentially and the following fusion operation to get $X^{fusion}$. In addition, a convolution layer with fixed parameters $Conv\_P$ and the global average pooling are added in parallel for $X^{img}$ to enable the accuracy of classification. Finally, the features $X^{fusion}$ and $X^{img}$ further go through the MIDN and a common softmax to set proposal scores and obtain the detection results through non-maximum suppression.

### 3.2 Pre-trained Backbone

As CNN can gradually extract features at different levels in the image (e.g. from edges and textures to semantic features of objects) by stacking convolutional and pooling layers, and pre-trained weights can speed up the learning of the network. Therefore, we built our method on the pre-trained CNN that has been trained on the ImageNet [23], and fine-tuned it on polyp data with only image-level supervision (i.e. no bounding box annotations) to get features $F = \{F_1, F_2, ..., F_N\}$ (where N is the number of feature blocks and N=5 in this study) and $Conv\_P$. We will give details of usage ($F$ & $Conv\_P$) in other Sections.

### 3.3 Cooperation-Based RPN

Since the purpose consistency between weakly supervised polyp detection and multi-instance learning, it is usually treated as MIL that requires previous proposals for the training data. Superpixel-based SSW is commonly used to generate the initial candidate boxes due to their ability to adapt to targets with blurred edges and does not require any annotation. However, they are rough because many of them have only the background or contain only a small percentage of polyps. Fortunately, SAM can generate accurate pseudo-masks for polyps to optimize the proposal but requires additional point prompts. In the classification task, pixels will respond according to their relevance with the target class, so there must be a pixel with the highest response in the object, and the coordinate of this point can exactly be used as a prompt for SAM. Therefore, we design the CBRPN to take advantage of both SSW and SAM to filter background noise as well as reduce incomplete polyp proposals before training to reduce the over-fitting risk.

In detail, the CBRPN takes the image I and feature $F_1$-$F_5$ as input. $I$ is used to generate an initial set proposals $B_{SSW} \in \mathcal{R}^{B_1 \times 4}$ ($B_1$: the number of proposals from SSW, 4: coordinates of each proposal for top, left, right and bottom) through SSW, and $F_1$-$F_5$ are used to produce the coordinate $(x_{point}, y_{point})$ for point according to the predicted category, formulated as below:

$$x\_point, y\_point = f_{u\_i}(argmax(ReLU(\sum_k \omega_k^c F^k))) \quad (1)$$

where $y\_point \& x\_point \in \mathcal{R}$, $F^k$ & $\omega_k^c$ denotes the $k_{th}$ feature maps and their weights corresponding to class c, $argmax(\cdot)$ is the operation to find the max pixel, $f_{u\_i}i$ is the unrval_index function

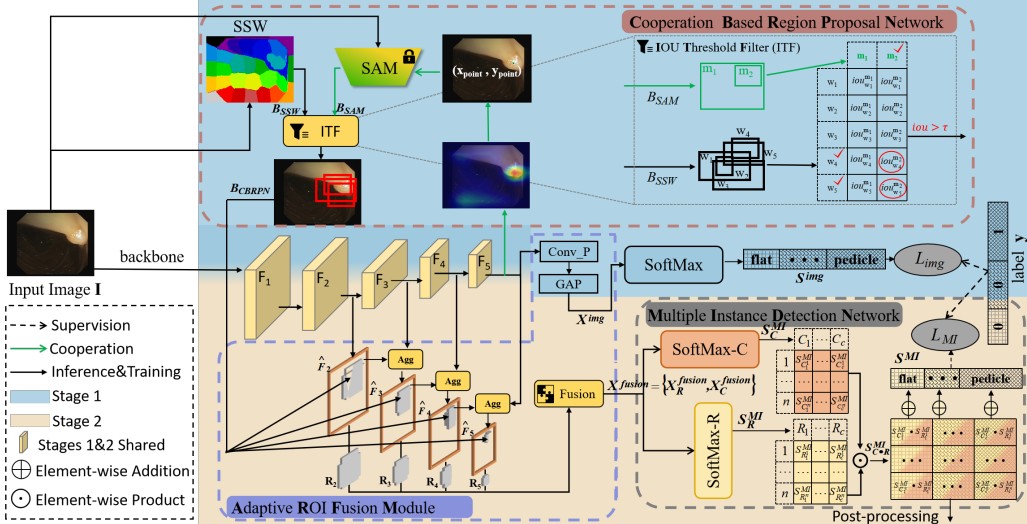

**Figure 2: Overview of the proposed CBNet with a two-stage training and testing strategy. Black arrows indicate feature flow. We use the Cooperation-Based Region Proposals Network (RPN) & Adaptive ROI (Region Of Interest) Fusion Model to generate high-quality proposals for training and enhance the ROI feature maps for accurate classifications. Multiple Instance Detection Network follows the same classification head as WSDDN for implementing the mapping of region scores to category scores.**

to find the coordinate of the max pixel. $\omega_k^c$ is defined as:

$$\omega_k^c = \frac{1}{Z} \sum_i \sum_j \frac{\partial y^c}{\partial F_{ij}^k} \tag{2}$$

where Z represents the number of pixels in the feature map, $\partial$ is the derivation operation. $y^c$ and $F_{ij}^k$ are the gradient of the $c_{th}$ score and the pixel value of the $K_{th}$ feature map at coordinates (i, j), respectively.

Since (x_point, y_point) can be regarded as the input prompt of the SAM, thereby another set of candidate boxes for filtering can be defined as:

$$B_{SAM} = f_{SAM}(I, x\_point, y\_point) \tag{3}$$

where $B_{SAM} \in \mathcal{R}^{B_2 \times 4}$ ($B_2$:number of proposals from SAM, 4:coordinates of each proposal for top, left, right and bottom). $f_{SAM}$ is the segment anything model with pre-training weights loaded. Thus, the significant noise can be filtered by ITF. The processes and rules can be formulated as:

$$B_{CBRPN} = \begin{cases} \kappa(m_i, w_j), if \frac{|m_i \cap w_j|}{|m_i| + |w_j|} > \tau \\ skip, otherwise \end{cases} \tag{4}$$

where $B_{CBRPN} \in \mathcal{R}^{B \times 4}$ (B: number of proposals after filtering, 4: coordinates of each proposal for top, left, right, and bottom), $m_i$ & $w_j$ are the proposals from $B_{SAM}$ and $B_{SSW}$, respectively. $\cap$ is the intersection of two sets, $|m_i \cap w_j|$, $|m_i|$ and $|w_j|$ represent the number of pixels in $m_i \cap w_j$, $m_i$ and $w_j$, respectively. $\tau$ is a threshold for the judgement of trade-offs, and $\kappa$ denotes the stack of proposals. Since the classification network may learn data bias [11] during training, the prompt point may appear in the background, which causes $B_{CBRPN}$ to be empty. When this occurs, Eq. 4 can be re-written as $B_{CBRPN} = B_{SSW}$.

By calculating the IOU between $B_{SSW}$ and $B_{SAM}$ and then removing proposals with low overlap through the ITF, the $B_{CBRPN}$ can suppress noise without polyps while strengthening the target learning by reducing false positives proposals.

## 3.4 Adaptive ROI Fusion Module

The ROI feature in the existing WS network typically is directly generated by the last convolution layer and then sent to region softmax to calculate the probability score. However, the features of this layer

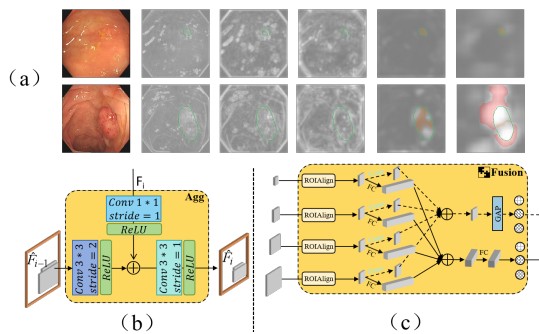

**Figure 3: Overview of the visualization feature maps (a) and the ARFM block. The ARFM contains an aggregation module to enhance extracted features and a fusion operation (b) to combine features at different levels.**

are limited and inaccurate, shown in Figure 3 (a), $2_{th}$ row Conv_5, which is inappropriate for medical images, e.g., the polyp images with a typical resolution of $224 \times 224$ in the morphological classification task. In addition, the flat polyp is usually small, and constant down-sampling will drown it in the background, and it is often

ignored due to the unclear boundary. To address the above problem, we develop an adaptive ROI fusion module (ARFM) inspired by the path aggregation mechanism (Refine-FPN) [18]. Specifically, ARFM has an agg and a fusion structure with detailed configuration inside as shown in Figure 3 (b), (c). We use $\{F_2, F_3, F_4, F_5\}$ to denote feature levels generated by the backbone. The aggregated mechanism starts from the lowest level $F_2$ and gradually approaches $F_5$. From $F_2$ to $F_5$, the spatial size is gradually being halved. We use $\{\hat{F}_2, \hat{F}_3, \hat{F}_4, \hat{F}_5\}$ to denote newly produced feature maps corresponding to $\{F_2, F_3, F_4, F_5\}$. Note that $\hat{F}_2 = F_2$, without any processing. The calculation process is as follows:

$$\hat{F}_i = ReLU(conv_{s=1}^{k=3}(f_1 + f_2))$$

$$f1 = ReLU(conv_{s=2}^{k=3}(\hat{F}_{i-1}))$$

$$f2 = ReLU(conv_{s=1}^{k=1}(F_i))$$

$$(5)$$

where $\hat{F}_i \in \mathcal{R}^{b \times C' \times H' \times W'}$ ($i = \{2, 3, 4, 5\}$ denote $i_{th}$ aggregation feature map, b is the batch_size, $C' = \{128, 256, 512, 512\}$, $H' = \{H/2, H/4, H/8, H/16\}$, $W' = \{W/2, W/4, W/8, W/16\}$), $conv_s^k$ is convolution operation with specific kernel k and stride s.

Further, we pool features from all levels for proposals to the same size $7 \times 7$ as:

$$\bar{R}_i = roi\_align(R_i) \qquad (6)$$

where $\bar{R}_i \in \mathcal{R}^{b \times 128 \times 7 \times 7}$ ( b: batch_size ), $R_i$ is the $i_{th}$ region feature cropped according to the proposal from $B_{CBRPN}$, roi_align is an adaptive pooling operation to adjust $R_i$ to a uniform size $7 \times 7$. Then, they are fused for the following prediction. Considering the influence of position information on subsequent modules, we design different fusion strategies: cnn-based and vector-based, detailed formulas are described as:

$$X_C^{fusion} = GAP(\sum_{i=2}^{5} conv_{s=1}^{k=3}(\bar{R}_i))$$

$$X_R^{fusion} = FC(\sum_{i=2}^{5} FC(\bar{R}_i))$$

$$(7)$$

where $X_C^{fusion}, X_R^{fusion} \in \mathcal{R}^{B \times C}$ (B: the number of proposals from $B_{CBRPN}$, C: the number of categories), GAP is the global average pooling (GAP), FC is the fully connected layer, $conv_s^k$ follow the same definition as Eq. 6.

Since ROI features only contain the local region of the image, this reduces classification accuracy and leads to over-fitting. To overcome this problem, we add the specific classification layer from the backbone to this module in parallel for global image feature $X^{img}$, which is calculated according to:

$$X^{img} = GAP(Conv\_P(F_5)) \qquad (8)$$

where $X^{img} \in \mathcal{R}^C$ (C is the number of morphological categories), $Conv\_P$ denotes the convolution that loads and freezes the pre-trained weights.

## 3.5 Multiple Instance Detection Network

Current WS object detection methods usually choose WSDDN as the criterion, which solves the problem that maps proposal scores at the instance level to image labels at the image level. Following the same double branch, we build a multiple instance detection

network (MIDN) to set scores for proposals. Specifically, MIDN consists of a location branch and a classification branch, the former selects which proposal region is more likely to contain the entire polyp fragment while the latter predicts which class to associate with the proposal region. Hence, the different scores of proposals can be represented:

$$S_C^{MI} = Softmax_C(X_C^{fusion}, dim = 1)$$

$$S_R^{MI} = Softmax_R(X_R^{fusion}, dim = 0)$$

$$(9)$$

where $S_C^{MI}, S_R^{MI} \in [0, 1]^{B \times C}$, $Softmax_C$ and $Softmax_R$ are both softmax operation that is responsible for mapping the feature matrix to the category dimension (dim=1) and the proposal dimension (dim=0), respectively. Finally, these score vectors are element-wise producted and added to obtain the image-level classification scores. The rule can be formulated as:

$$S_{C \bullet R}^{MI} = S_{C_j^i}^{MI} \bullet S_{R_j^i}^{MI}$$

$$S^{MI} = \sum_{r=1}^{B} S_{C \bullet R}^{MI}$$

$$(10)$$

where $S_{C \bullet R}^{MI} \in [0, 1]^{B \times C}$, $S^{MI} \in [0, 1]^C$. Additionally, the feature $X^{img}$ of the whole image also be fed to a softmax to get the image-level $S^{img} \in [0, 1]^C$:

$$S^{img} = Softmax(X^{img}, dim = 1) \qquad (11)$$

## 3.6 Loss Function

**BCE Loss**: Binary Cross Entropy Loss is a measure used to evaluate the distance between the prediction and image label. We employ it as the loss function to train our network:

$$L = L_{img} + L_{MI}$$

$$L_{img} = -\frac{1}{N}\sum_{i=1}^{N} y_i \bullet log(p(S_i^{img})) + (1 - y_i) \bullet log(1 - p(S_i^{img}))$$

$$L_{MI} = -\frac{1}{N}\sum_{i=1}^{N} y_i \bullet log(p(S_i^{MI})) + (1 - y_i) \bullet log(1 - p(S_i^{MI}))$$

$$(12)$$

where N indicates the number of predicted object groups, $y_i$ is the one-hot label of $i_{th}$ category, $p(S_i^{img})$ and $p(S_i^{MI})$ are the probabilities belong to $i_{th}$ class predicted by model.

## 4 Experiments

### 4.1 Datasets

**CVC-ClinicDB**[1] [3]: The dataset comprises 612 images sourced from 29 colonoscopy video sequences, each with a resolution of $288 \times 384$. It was developed in partnership with the Hospital Clinic of Barcelona, Spain, and consists of 322 flat, 282 pedicle, and 66 edge polyps.

**Kvasir**[2] [9]: This dataset was assembled by Vestre Viken Health Trust in Norway which includes 1,000 polyp images along with their ground truth annotations from colonoscopy videos. The bounding boxes for ground truth were initially

---

[1]https://polyp.grand-challenge.org/CVCClinicDB/
[2]https://datasets.simula.no/downloads/kvasir-seg.zip

marked by medical doctors and later confirmed by experienced gastroenterologists. The images span a range of resolutions from 332 × 487 to 1920 × 1072 pixels. The number of three morphologies is 255, 851, and 53, respectively.

**Private**: Our internal dataset extracted 290 static images from OlympiusEurope colonoscopy videos at a hospital, which consists of 177 patients, annotated and validated by experienced endoscopists. We rejected all the images with extremely high patient preparation or bad visualization quality due to image blurring and each image in the dataset is paired with a morphological image label (87 flat, 158 pedicle, 45 edge) to guide the training. The images vary in resolution spanning from 564×480 to 600×530 pixels. To assess the location of polyps, experts also provided ground truth (bounding boxes) for all images. In addition, to keep the dataset up-to-date we are still continuously collecting updated data.

### 4.2    Implementation Details

**Training Details.** The slightly modified classifier layer of VGG16 serves as our backbone. Specifically, we replace the classifier of three FC operations with a GAP layer and reshape after the original last feature layer. In the first stage, we only train the backbone for a total of 25 epochs with a learning rate of 1e-4 and a batch size of 32. And the trained weight sam_vit_b_01ec64 [3] on nature images is used as the parameter of SAM. In the second stage, we train the CBNet for a weight decay of 5e-4, a batch size of 1, and 15 epochs with a learning rate 1e-5 following 5 epochs with 1e-6. As discussed in Eq 4, we empirically set $\tau$ as 0.5. We divided data into two splits on each dataset: training, and test. The training split comprises about 80% of data; the test about 20% each.

**Evaluation Metric.** To evaluate detection performance, we employ three performance measures. The first one follows the standard PASCAL VOC protocol, calculating average precision (AP) and mean AP (mAP) at IOU thresholds of 10%, 30%, 50% between the detected boxes and the ground truth ones. Additionally, we report CorLoc for location, a commonly used weakly supervised detection measure [10], which means the percentage of images where at least one instance of the target object class is correctly localized with the most confident detected bounding box overlapping at least 50%. Detailed explanations of all evaluation indicators can be found in the cited references.

### 4.3    Quantitative Results

**Compared Methods**: Some well-performed detection methods are selected for comparison. They are categorized into fully and weakly supervision (Fully sup. and Weakly sup.). methods. In the fully sup. methods, we choose **Faster**

**Rcnn** [28], **Yolo** [27] and **Diffusion** [6] with dynamic boxes 50 & 500. In the weak sup. methods, we choose **WSDDN** [13], **OICR** [32], **WSOD2** [40], **Grad-CAM** [29], **Grad-CAM++** [2], IDC [34] and LPCAM [7]. Unfortunately, the last two methods don't work very well, so we only list the results of other methods. The results of Faster RCNN, Yolo, Diffusion-Det are given from mmdetection[4] [5], WSOD2 are referred from codes [5] and the rest are implemented by ourselves with source codes.

**Main Results**: For a clearer comparison, we represent the best indicator of CBNet as red and the best indicator of weak supervision as blue. As shown in Table 1, traditional weakly supervised methods have a limited ability to deal with challenging polyps, thereby having much worse scores than fully supervised methods and ours. For example, the best mAP of weakly sup. is only 0.3 (IOU@10-50) on CVC-ClinicDB but full supervision can reach 0.6. Even better, our method reaches 0.64, which not only exceeds weak supervision or even due to full supervision. In comparison with CVC-ClinicDB and private dataset, the best mAP (IOU@10) on Kvasir is only 0.22, which is 55% lower than CVC-ClinicDB, and 30% lower than private. Besides, we also find that although the full-supervision methods have the high CorLoc, mAP does not have the same advantages, e.g. 1.0 CorLoc v.s. 0.39 mAP (IOU@30) on the private of DiffusionDet, which indicates that they are inefficient in learning discriminative category representation, leading to false detection or missed detection. Compared with them, our proposed CBNet offers higher mAP despite having slightly low CorLoc due to the lack of instance-level annotations, which shows that CBNet can learn a better target representation to deal with flat polyps and background noise issues. More detail for each category can be seen in **supplementary material**.

**Precision-Recall Curves**: We also plot the Precision-Recall (P-R) curves of different methods on the CVC-ClinicDB dataset and private dataset for flat polyps in Figure 4. As can

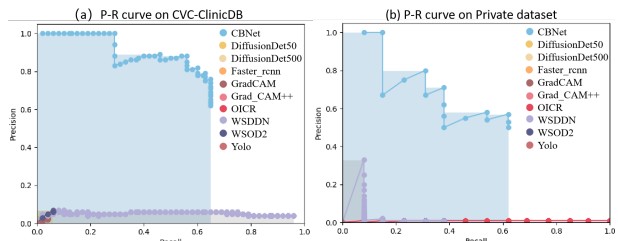

**Figure 4: P-R curves of different methods on the CVC-ClinicDB (a) and private dataset (b) for flat polyps.**

be seen, the performance of our CBNet is significantly better than all other methods, where the area under the P-R

---

[3]https://dl.fbaipublicfiles.com/segment_anything/

[4]https://github.com/open-mmlab/mmdetection
[5]https://github.com/researchmm/WSOD2

**Table 1: Quantitative comparisons with eight methods on three polyp datasets in terms of mAP and CorLoc from different iou thresholds. The top two results in weakly sup. methods are marked in red and blue font, respectively.**

| Methods | Supervision | CVC-ClinicDB | | | | | | | Kvasir | | | | | | | Private | | | | | | |
|---|---|---|---|---|---|---|---|---|---|---|---|---|---|---|---|---|---|---|---|---|---|---|
| | | IoU@10-50 | IoU@10 | | IoU@30 | | IoU@50 | | IoU@10-50 | IoU@10 | | IoU@30 | | IoU@50 | | IoU@10-50 | IoU@10 | | IoU@30 | | IoU@50 | |
| | | mAP | mAP | CorLoc | mAP | CorLoc | mAP | CorLoc | mAP | mAP | CorLoc | mAP | CorLoc | mAP | CorLoc | mAP | mAP | CorLoc | mAP | CorLoc | mAP | CorLoc |
| Faster Rcnn [28] | Fully Sup. | 0.41 | 0.41 | 0.94 | 0.41 | 0.91 | 0.41 | 0.90 | 0.26 | 0.27 | 0.96 | 0.26 | 0.95 | 0.26 | 0.92 | 0.28 | 0.28 | 0.97 | 0.28 | 0.97 | 0.28 | 0.94 |
| Yolo [27] | Fully Sup. | 0.40 | 0.41 | 0.99 | 0.40 | 0.96 | 0.39 | 0.89 | 0.27 | 0.28 | 0.95 | 0.27 | 0.93 | 0.26 | 0.88 | 0.29 | 0.30 | 0.84 | 0.30 | 0.76 | 0.28 | 0.74 |
| DiffusionDet50 [6] | Fully Sup. | 0.62 | 0.63 | 0.97 | 0.63 | 0.97 | 0.59 | 0.96 | 0.28 | 0.29 | 0.97 | 0.28 | 0.96 | 0.26 | 0.94 | 0.37 | 0.38 | 1.00 | 0.38 | 1.00 | 0.37 | 1.00 |
| DiffusionDet500 [6] | Fully Sup. | 0.61 | 0.62 | 0.98 | 0.62 | 0.98 | 0.59 | 0.97 | 0.26 | 0.27 | 0.96 | 0.27 | 0.94 | 0.24 | 0.89 | 0.38 | 0.40 | 1.00 | 0.39 | 1.00 | 0.34 | 1.00 |
| WSDDN [13] | Weakly Sup. | 0.11 | 0.17 | 0.35 | 0.09 | 0.14 | 0.07 | 0.05 | 0.06 | 0.08 | 0.37 | 0.06 | 0.13 | 0.05 | 0.05 | 0.12 | 0.15 | 0.24 | 0.12 | 0.08 | 0.11 | 0.04 |
| OICR [32] | Weakly Sup. | 0.02 | 0.04 | 0.09 | 0.00 | 0.02 | 0.00 | 0.02 | - | - | - | - | - | - | - | 0.01 | 0.01 | 0.05 | 0.01 | 0.04 | 0.01 | 0.04 |
| WSOD2 [40] | Weakly Sup. | 0.30 | 0.38 | 0.59 | 0.26 | 0.36 | 0.25 | 0.32 | 0.02 | 0.02 | 0.10 | 0.02 | 0.08 | 0.02 | 0.07 | 0.01 | 0.01 | 0.04 | 0.01 | 0.04 | 0.01 | 0.04 |
| Grad-CAM [29] | Weakly Sup. | 0.20 | 0.42 | 0.51 | 0.13 | 0.24 | 0.04 | 0.08 | 0.06 | 0.12 | 0.37 | 0.04 | 0.16 | 0.01 | 0.06 | 0.16 | 0.24 | 0.31 | 0.13 | 0.20 | 0.10 | 0.17 |
| Grad-CAM++ [2] | Weakly Sup. | 0.22 | 0.49 | 0.54 | 0.13 | 0.24 | 0.04 | 0.08 | 0.06 | 0.12 | 0.36 | 0.04 | 0.17 | 0.01 | 0.08 | 0.15 | 0.22 | 0.28 | 0.13 | 0.19 | 0.10 | 0.15 |
| CBNet(SAM&SSW) | Weakly Sup. | 0.64 | 0.77 | 0.93 | 0.62 | 0.85 | 0.52 | 0.74 | 0.16 | 0.20 | 0.95 | 0.16 | 0.80 | 0.11 | 0.64 | 0.49 | 0.52 | 0.90 | 0.49 | 0.88 | 0.48 | 0.84 |
| CBNet(SAM(filter)&SSW) | Weakly Sup. | 0.51 | 0.53 | 0.91 | 0.52 | 0.88 | 0.49 | 0.81 | 0.18 | 0.22 | 0.91 | 0.18 | 0.79 | 0.13 | 0.63 | 0.49 | 0.51 | 0.88 | 0.48 | 0.84 | 0.47 | 0.82 |

curve of our CBNet is much larger than those of both the fully supervised methods and weakly supervised methods, e.g. 0.6029 area of CBNet v.s. 0.0565 area of WSDDN on the CVC-ClinicDB dataset.

## 4.4 Visual Results

In Figure 5, we present some visual detection results of different methods. As we can see, the $9_{th}$ column test image contains a flat polyp, all the methods including both fully supervised and weakly methods falsely detect or miss the polyp as a target except our CBNet, which shows that our CBNet has more excellent representation ability in complex flat polyps. We attribute this success to the unique structure of our ARFM, i.e., it aggregates different level features such that it has the ability to provide more information for following modules and recognize polyp as target rather than background. Moreover, the last two rows show that overall our method has better performance, and different proposal filtering strategies will make a negative or positive impact on the results.

To gain more in-depth in-sight into what backbone has learned, we also visualize the gradient-weighted class activation mapping (Grad-CAM), as in Figure 6. It can be observed that baseline has the ability to successfully locate polyps but only focus on a small part of the target e.g., $1_{th}$ row & $2_{th}$ column, and may occasionally fail e.g., $1_{th}$ row & $1_{th}$ column, $2_{th}$ row & $3_{th}$ row.

## 4.5 Ablation Study

Considering that the detection of backbone is CAM-based and has a nature performance gap (e.g. the performance of WSDDN v.s. Grad-CAM in Table 1) with the MIL-based method, we selected WSDDN as the reference in order to ensure the fairness of comparison.

**Impact of CBRPN:** To investigate the impact of the proposed CBRPN, we conduct ablation studies by using SAM's proposal as a result, directly using proposals from SAM and filtering them based on the area before using. The results are summarized in Table 2. As can be seen, compared to other proposal generation methods, CBRPN achieves higher mAP. And the lower flat AP is due to insufficient learning ability

**Table 2: The impact of CBRPN with different proposal generation strategies on Kvasir.**

| Proposal Source | mAP(%) | Flat AP(%) | Pedicle AP(%) | Edge AP(%) |
|---|---|---|---|---|
| SSW | 4.71 | 2.79 | 11.36 | - |
| SAM | 5.55 | - | 16.64 | - |
| SAM(filter) | 7.29 | - | 21.86 | - |
| SAM&SSW | 10.38 | 1 | 30.16 | - |
| SAM(filter)&SSW | 10.84 | 0.18 | 26.45 | 5.9 |

**Table 3: Performance of our ablation experiments for ARFM with different strategy CBRPN.**

| Strategies | Datasets | mAP(%) | Flat AP(%) | Pedicle AP(%) | Edge AP(%) |
|---|---|---|---|---|---|
| 1) SSW | CVC-ClinicDB | 6.69 | 5.65 | 11.45 | 2.99 |
| | Kvasir | 4.71 | 2.79 | 11.36 | - |
| 2) SSW+ARFM | CVC-ClinicDB | 35.54 | 50.37 | 20.68 | 35.38 |
| | Kvasir | 7.18 | 4.88 | 16.65 | - |
| 3) SAM&SSW+ARFM | CVC-ClinicDB | 51.94 | 77.29 | 42.97 | 35.58 |
| | Kvasir | 11.02 | 6.58 | 26.48 | - |
| 4) SAM(filter)&SSW+ARFM | CVC-ClinicDB | 49.17 | 63.88 | 45.15 | 38.46 |
| | Kvasir | 13.44 | 14.25 | 18.33 | 7.74 |

in the subsequent modules, as demonstrated in ablation experiments of the ARFM module. ' - ' represents 0.0, and the possible reasons for this result are as follows: 1) more negative samples (compared with those related to SSW) which affect the learning of positive samples; 2) backbone network lacks the ability to learn small differences among classes.

In addition, we also compared the average number of proposals used for training and the average overlap with ground truth boxes. The detailed results are in **supplementary material**.

**Impact of the design of ARFM:** We investigate the effect of ARFM on the original reference network as well as different proposal filtering strategies. The results on CVC-ClinicDB and Kvasir are shown in Table 3. Comparing strategy 1 v.s. 2, we can find that the addition of ARFM contributes to a gain of about 28.85% mAP for CVC-ClinicDB and 2.47% mAP for Kvasir, as well as increasing the AP of each morphology in different degree. Again, from the $3_{th}$ strategy to $4_{th}$ strategy ARFM outperforms $1_{th}$ & $2_{th}$ strategy in all metrics. Finally, comparing strategies 3 and 4 we find that SAM self-filtering based on the area can further improve performance on Kvasir, but is negative on CVC-ClinicDB.

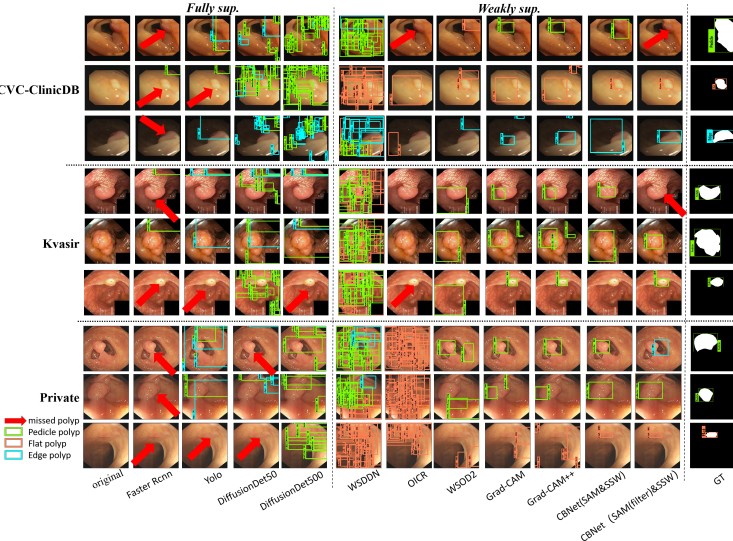

**Figure 5: Qualitative comparison with different methods. Rows 2 to 5 are fully supervised methods, the others are weakly supervised methods.**

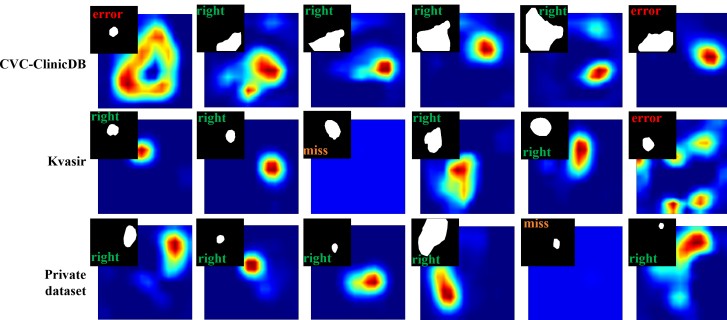

**Figure 6: The grad class activation maps of backbone, which include three location case** error, right **and** miss.

**Table 4: The evaluation results of each component.**

| Module | Strategies | Datasets | mAP(%) | Module | Strategies | Datasets | mAP(%) |
|---|---|---|---|---|---|---|---|
| CBRPN | w/ SSW | CVC-ClinicDB | 6.69 ↓7.14 | ARFM | w/ CG | CVC-ClinicDB | \ |
| | | Kvasir | 4.71 ↓5.67 | | | Kvasir | \ |
| | w/ SAM | CVC-ClinicDB | 8.33 ↓5.5 | | w/ AF | CVC-ClinicDB | 49.95 ↓1.99 |
| | | Kvasir | 5.55 ↓4.83 | | | Kvasir | 9.88 ↓1.14 |
| | w/ (SSW + SAM) | CVC-ClinicDB | **13.83** | | w/ (CG + AF) | CVC-ClinicDB | **51.94** |
| | | Kvasir | **10.38** | | | Kvasir | **11.02** |

The probable reason for this is that the latter has many small polyps and the inappropriate area threshold filters them out. In a word, the proposed ARFM can help the network achieve better performance in all metrics.

**Impact of the components of CBRPN & ARFM:** We further performed ablation experiments on the components within each module to verify their role for the module. The results of CBRPN {*SSW*, *SAM*} and ARFM {*CG*: *Conv_P + GAP*, *AF*: *Agg + Fusion*} are reported in Table 4. ' / 'indicates that it cannot be evaluated because the input of subsequent modules is the output of AF. We can find that all components are beneficial to our framework because the performance

decreases (4.83 ~ 7.14 for CBRPN, 1.14 ~ 1.99 for ARFM) while removing each component.

## 5 Conclusion

In this paper, we propose a novel CBNet for image-level weakly supervised polyp detection. CBNet adopts CBRPN to automatically generate point prompts for SAM taking advantage of the local properties in the classification and adds iou threshold filter to improve the quality of proposals. We also propose the ARFM to enhance the region feature, which further helps to detect flat polyps and avoid over-fitting. Experiments on public datasets and internal demonstrate the superiority of our CBNet.

## Acknowledgments

The work was supported by Hefei Municipal Natural Science Foundation (2022009) and the High-performance Computing Platform of Anhui University.

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
