# OpenReview forum: "CBNet: Cooperation-Based Weakly Supervised Polyp Detection"
_acmmm.org/ACMMM/2024/Conference — MM2024 Poster_

### Official Review · Reviewer_Pxxt · 2024-05-22

**Rating:** 4
**Confidence:** 3

**Summary:**

This paper introduces a Cooperation-Based network (CBNet) for polyp detection, supervised by image labels, which operates in two stages. The model combines classification and segmentation to eliminate wrong proposals and achieve more accurate detection results by aggregating multi-level roi features. The proposed model aims to address several issues present in existing research: (1) over-fitting polyp detection; (2)unfilterable noise; (3)failed polyp mask without point annotation; (4)missed flat polyps. Moreover, experimental validation on three datasets demonstrates the excellent performance of the proposed model in the weakly supervised polyp detection task. Further, through ablation and visualization experiments, the effectiveness of each component of the model is further validated.

**Strengths:**

1. The authors proposed a comprehensive model to address the challenges faced by existing methods. As for the CBRPN module, it effectively reduces the negative impact of noise and ensure the quality of generated proposals, thus improving the network's ability to capture relevant features.
2. The ARFM module’s ability to aggregate multi-level ROI features enhances the accuracy and integrity of polyp detection. This addresses the challenge of detecting small and flat polyps that are often missed.
3. The paper includes extensive experiments on both public and private datasets, demonstrating the robustness and generalizability of the proposed method. And it achieves state-of-the-art performance for weakly supervised methods and, in some metrics, outperforms fully supervised approaches, showcasing its practical utility in clinical settings.

**Limitations:**

1. There is still inconsistency between the text and the figures in the article. For instance, the description of the convolution operation in Equation 5 on page four is inconsistent with Fig. 3.
2. The ablation experiments only focused on the removal of each module. It would be more persuasive if further evaluation could be conducted to assess the roles played by individual components within each module.
3. The proposed method involves multiple stages and components, which may increase the computational complexity. Therefore, an analysis of its training time and model complexity is expected.

**Suitability:**

2

---

### Official Review · Reviewer_Rbzo · 2024-05-22

**Rating:** 4
**Confidence:** 3

**Summary:**

This paper proposes a novel cooperation-based weakly supervised polyp detection method (CBNet). Based on the four challenges facing the weakly supervised polyp detection model, two innovative modules are proposed correspondingly. In which the cooperation-based regional proposal network calculates IOU through proposals obtained from the proposal generator and masks obtained from SAM, leaving relatively complete information. And, the adaptive ROI fusion module combines deep-shallow region features to improve the model's ability to detect flat polyps. Extensive experiments on three polyp datasets demonstrate the effectiveness and superiority of the proposed method.

**Strengths:**

(1) This paper is well-written and well-organized. The motivation of the paper is quite clear.
(2) The proposed methodology is innovative. The cooperation-based regional proposal network is a promising idea. This method can filter background noise and reduce the risk of overfitting could be recognized.
(3) The experiment is quite thorough. Extensive experiments in the paper demonstrate that the proposed method achieves state-of-the-art performance. The proposed method shows higher accuracy compared to other weakly supervised methods and even higher than fully supervised methods.

**Limitations:**

(1) Missing statement of hyperparameters. I would like to know how the hyperparameters (e.g. $\tau$) are determined. Are there any papers referenced or relevant ablation experiments performed?
(2) Lack of comparison with the latest methods. This paper does show extensive experiments. However, the comparison with the latest weakly supervised detection method (published in 2022 or 2023) is missing, and thus the superiority of the proposed method cannot be fully proved.
(3) Inadequate explanation of some experiments. The paper lacks the explanation of why the mAP and CorLoc of the CBIA-WSPD (samAndssw) and CBIA-WSPD (samSelfFilterAndssw) methods differ more on the Kvasir dataset relative to the other datasets.
(4) MINOR: Some typos need to be corrected, i.e. 'Kvasir' is written as 'Kavsir' in Table 1.

**Suitability:**

2

---

### Official Review · Reviewer_EypZ · 2024-05-25

**Rating:** 4
**Confidence:** 4

**Summary:**

This paper introduces a novel deep learning framework, termed CBNet, for Weakly Supervised Polyp Detection. The proposed method contains two innovative modules: Cooperation-Based Region Proposal Network (CBRPN) and Adaptive ROI Fusion Module (ARFM). The proposed method is interesting enough. However, the writing quality of this paper is not good. The authors should revise it for the camera-ready version.

**Strengths:**

1) The code is available.
2) A comprehensive set of experiments in three datasets have been conducted.
3) The proposed method is theoretically sound, and aligned with the problem formulation and the motivation.

**Limitations:**

1) The writing quality of this paper is not particularly good. Some sentences need to be clarify:

In lines 111-113, the authors point out that the MIL-based and SAM-aided approaches have complementary advantages and disadvantages. However, the authors do not mention about the disadvantages.

In line 135-138, the authors mention about "the former" and "the latter". "The former" and "the latter" mean what?

What is SSW. In the related work section, authors only mention about the Selective Search (SS).

In line 370-374, this sentence could be rewritten for better clarity. the CBRPN takes the image I and feature F_1 - F_5 as inputs. The image I is used to generate the B_SSW and the feature F_1 - F_5 are used to produce the coordinates.

The authors should mention the WSDDN in the related work section. Since the MIDN is built upon the WSDDN.

The caption of Table 1 need to be rewrite. Since the best indicator of CBNet is red font and the best indicator of weak supervision is blue font. Besides, What is CBIA-WSPD in Table 1?

2) In Table 2, why SAM and SAM(filter) do not have results?

**Suitability:**

2

---

### Official Review · Reviewer_4Frq · 2024-06-04

**Rating:** 5
**Confidence:** 3

**Summary:**

The authors introduce a novel Cooperation-Based Network (CBNet), a two-stage framework for polyp detection that is supervised by image labels. In the proposal generation stage, they present a Cooperation-Based Region Proposal Network (CBRPN) that leverages the collaboration of classification and segmentation to filter noise and remove incomplete polyp proposals, thereby mitigating the risk of overfitting. Additionally, they design an Adaptive ROI Fusion Module (ARFM) to enhance the detection of flat polyps by fusing multi-level region features. Extensive experiments demonstrate the effectiveness of the proposed method, achieving state-of-the-art performance on both public and private datasets.

**Strengths:**

1. The authors effectively identify and address key challenges in current weakly supervised polyp detection approaches, including over-fitting, unfilterable noise, failed polyp masks without point annotation, and missed flat polyps. Their analysis and observations provide a solid foundation for the proposed model.
2. The adoption of the Segment Anything Model (SAM) to obtain relatively complete polyp proposals and filter out noise is both novel and effective.
3. The proposed method demonstrates a significant improvement over compared approaches. Extensive experiments show that the performance of CBNet not only surpasses other weakly supervised methods but also, in some respects, exceeds fully supervised methods, showcasing its effectiveness and potential impact in the field.

**Limitations:**

1.The comparison approaches in the experimental section are somewhat outdated. Incorporating more recent state-of-the-art approaches would provide a stronger benchmark and better highlight the improvements and relevance of the proposed method.

**Suitability:**

2

---

### Meta-Review · Area_Chair_1WNV · 2024-07-02

**Recommendation:** Accept (Poster)
**Confidence:** 4

**Metareview:**

Overall, all reviewers are satisfied with the response given by the authors, and are glad to see that the quality of the paper has been improved substantially.